# Cardiotonic Steroids—A Possible Link Between High-Salt Diet and Organ Damage

**DOI:** 10.3390/ijms20030590

**Published:** 2019-01-30

**Authors:** Aneta Paczula, Andrzej Wiecek, Grzegorz Piecha

**Affiliations:** Department of Nephrology, Transplantation and Internal Medicine, Medical University of Silesia, Francuska 20-24, 40-027 Katowice, Poland; paczula@poczta.onet.pl (A.P.); awiecek@sum.edu.pl (A.W.)

**Keywords:** marinobufagenin, ouabain, salt, hypertension, fibrosis

## Abstract

High dietary salt intake has been listed among the top ten risk factors for disability-adjusted life years. We discuss the role of endogenous cardiotonic steroids in mediating the dietary salt-induced hypertension and organ damage.

## 1. Salt and Blood Pressure

High dietary salt intake has been listed among the top ten risk factors for disability-adjusted life years (DALYs) [1]. The role of dietary salt in the pathogenesis of increased blood pressure has been demonstrated by several large clinical trials, such as the International Study of Salt and Blood Pressure (INTERSALT) [2] and the Dietary Approaches to Stop Hypertension (DASH) study [3]. High salt consumption is associated with increased blood pressure (BP) and vascular stiffening due to altered endothelial and vascular smooth muscle cells (VSMCs) function and extended arterial wall fibrosis [4,5,6]. Notably, high dietary salt intake correlates positively with a faster pulse wave velocity (PWV), indicating arterial stiffening, which precedes the development of hypertension with aging [7,8,9]. Conversely, dietary salt restriction is accompanied by a reduction in PWV, indicating less arterial stiffening [10]. 

Salt/sodium is absolutely necessary for survival. As the availability of salt/sodium is scarce in nature outside the oceans, the mechanisms for salt conservation are very efficient and well known. However, the mechanisms for elimination of excess salt are less understood. 

Laragh et al. postulated two forms of essential hypertension: one related to vasoconstriction (largely the result of the renin-angiotensin system activation) and the other form due to volume expansion (excess salt and water) in which plasma renin activity is suppressed [11].

An unresolved issue in the pathogenesis of hypertension is the specific mechanism or “signaling pathway” by which salt retention elevates the blood pressure (BP). Mean arterial BP is a function of cardiac output (CO) and total peripheral vascular resistance (TPR). Cardiac output, which is generated by a heart rate (HR) and stroke volume (SV), is in turn directly related to the extracellular fluid volume, specifically the volume of the venous return to the heart. TPR is regulated by vasoconstriction or vasodilatation of small resistance arteries by three mechanisms: baroreflexes and other neuro-humoral mechanisms, endothelial and myogenic mechanisms. Hypertension has often been associated with structural changes in arterial wall that decrease the wall-to-lumen ratio and increase wall stiffness. It is not clear, however, whether such a vascular remodeling is only a consequence of hypertension or is it also an important factor in the pathogenesis of elevated blood pressure. Recently, it was reported that in some models of hypertension, most of the increase in TPR can be attributed to functional and not structural alterations in small resistance arteries. The contraction of the vascular smooth muscle cells (VSMCs) is activated by a rise in the cytosolic Ca^2+^ concentration [12].

For many years it has been assumed that increased sodium intake is paralleled by increased sodium excretion maintaining steady sodium body content and that sodium is accumulated only with a corresponding volume of extracellular fluid. This assumption places the kidney as a central regulator of sodium handling. In recent years, sodium has been found to be accumulated in osmotic inactive state in the interstitium of the skin and other organs [13]. Regulation or dysregulation of this storage may affect blood pressure. Some data suggests that sodium and potassium may regulate the stiffness of endothelial cells and their nitric oxide release and thus the vessel tone and blood pressure [14]. Central nervous system emerged as another site of salt sensing in cerebrospinal fluid by a novel isoform of Na channels (Na_x_), sensing of CSF osmolality by nonselective cation channels (transient receptor potential vanilloid type 1 channels), and osmolarity sensing by volume-regulated anion channels in glial cells of supraoptic and paraventricular nuclei [15].

## 2. “Humoral Factor” Increases Blood Pressure in Response to Salt Intake

The hypothesis of a circulating “humoral factor” that induces salt-sensitive hypertension came from the study performed by Dahl et al. over half a century ago [16,17]. Later, de Wardener and Clarkson suggested that this unidentified “humoral factor”, implicated in the pathogenesis of salt-sensitive hypertension, was an endogenous natriuretic hormone, and had digitalis-like properties [18]. Cardiotonic steroids (CTS, Figure 1) were first found in plants, most notably digitalis in the foxglove plant, and then in the skin of toads like the *Bufo marinus* [19]. They have been used in traditional ancient medicine to treat congestive heart failure [20]. Endogenous CTS have been implicated in sodium homeostasis and blood pressure regulation through their effects on the Na/K-ATPase in renal and cardiovascular tissue [19]. Cardiotonic steroids (CTS) are also called digitalis-like factors. They are a group of steroid hormones that circulate in the blood and are excreted in the urine. CTS synthesis has been demonstrated in the adrenal cortex, placenta and hypothalamus [21]. They belong to two groups with different chemical structure: cardenolides (e.g., ouabain) and bufadienolides (e.g., marinobufagenin). Until recently, their biological role has been linked to their ability to inhibit activity of the ubiquitous transport enzyme called sodium-potassium adenosine triphosphatase (Na/K-ATPase). Over the last several years, their signaling capabilities unrelated to the Na/K-ATPase inhibition have caught the attention of many scientific groups.

## 3. Na/K-ATPase: A Pump and a Receptor

The Na/K-ATPase is an ubiquitous enzyme present on the surface of all cells, the primary role of which is to maintain the difference in natrium and potassium concentrations between cytosolic and extracellular compartments. These differences are essential for cell-to-cell communication, contractility, and response to stimuli. The Na/K-ATPase is a heterodimer consisting of alpha and beta subunits. The alpha subunit is the “catalytic subunit” and contains binding sites for ATP, CTS, and other ligands, while the beta subunit is essential for the structural assembly of the enzyme. There are four α and three β isoforms known, thus allowing numerous combinations of αβ complexes among tissues with different characteristics including different sensitivity to different cardiotonic steroids. The α1β1 complex is the most common combination and is present in nearly every tissue. The α2 isoform is expressed in adult heart, smooth muscle, skeletal muscle, brain, adipocytes, cartilage, and bone. The α3 isoform is expressed in the central and peripheral nervous tissues and in the conductive system of the heart. The α4 isoform has been found only in testis. The β2 and β3 isoforms are expressed in the brain, cartilage and erythrocytes, whereas β2 can also be found in cardiac tissue and β3 in lungs. The cardenolides have been determined to have a predilection for the α2 and α3 isoforms (Table 1 [22,23]), whereas the bufadienolides also inhibit the α1 isoform. There are, however, differences between species in terms of the sensitivity of those isoforms to different CTS (e.g., in rats the α1 isoform is resistant to ouabain, while in humans it is not). The K_i_ values of human α1, α2 and α3 isoforms range from 10^−8^ to 10^−9^ M/l [23]. Differences have even been found in different cellular localization of the enzyme: the α1 Na^+^/K^+^-ATPase, expressed in the renal epithelium, is ouabain-resistant, while the α1 isoform, found in the caveolae of renal tubular cells, exhibits remarkable sensitivity to ouabain [24]. In *rats* and in *humans*, ouabain has been detected in plasma at concentrations between 10^−9^ and 10^−10^ M/l [25]. Marinobufagenin has been reported in *rat* plasma at concentrations 10^−9^ to 10^−10^ M/l and in *human* plasma at concentrations between 0.5 × 10^−9^ and 10^−8^ M/l [25].

Apart from the “classic” function of the Na/K-ATPase of maintaining the gradient of sodium and potassium concentrations across the plasmalemmal barrier, an alternative or “signaling” function for the enzyme has been described in recent years. This model proposes that some of plasmalemmal Na/K-ATPase resides in the caveole of the cells and does not seem to actively “pump” sodium and potassium but is closely associated with other key signaling proteins [19]. The Na/K-ATPase has been colocalized with signaling molecules including Src, PLC-γ, PI3K, IP3R, ankyrin, adducin, and caveolin-1 [26].

Activation of this receptor complex by CTS results in stimulation of the protein kinase cascades and generation of second messengers. Binding of ouabain to the caveolar complex of Na/K-ATPase phosphorylates epithelial growth factor receptor (EGFR) via Src and this results in activation of the Ras/Raf/MEK/ERK1/2 cascade [27]. These ouabain-induced signaling events may be specific for a particular cell type. For example, ouabain simulates the Src-dependent activation and translocation of several PKC isoforms in cardiac myocytes, which in turn activate the Ras/Raf/ERK1/2 cascade [28]. Moreover, in cardiac myocytes ouabain is also able to induce phosphorylation of protein kinase B (Akt) [29]. The cumulative effects of Akt, ERK1/2 and calcium signaling results in hypertrophic growth of cardiac myocytes, stimulate proliferation in renal epithelial cells [29], but cause growth inhibition in some cancer cells [30].

## 4. Marinobufagenin is a Ligand for Na/K ATPase

Marinobufagenin (MBG) by inhibiting the Na/K-ATPase participates in the regulation of renal sodium transport and arterial blood pressure. MBG promotes natriuresis through inhibition of sodium pump in the renal proximal tubules and vasoconstriction through inhibition of the same enzyme in vascular smooth muscle cells [31]. The synthesis of MBG by the adrenocortical cells is stimulated by high salt intake and is observed in volume-expanded states, such as preeclampsia, chronic kidney disease, and resistant arterial hypertension [31]. Elevation of plasma MBG concentration is preceded by transient ouabain increase [32]. Ouabain does not have natriuretic properties and increase of its plasma concentration after increased salt intake is only short-term. Inhibition of Na/K-ATPase in the vascular smooth muscle cells (VSMC) results in an increase of cytosolic Ca^2+^ concentration through Na^+^/Ca^2+^ exchanger (NCX) which results in VSMC contraction [33]. 

Only some of the endogenous cardiotonic steroids may increase natriuresis. This is not only due to inhibition of the Na/K-ATPase and subsequently renal sodium reabsorption, but also due to internalization of the sodium pump in the proximal tubule and decreased expression of the transport protein, Na/H exchanger (NHE3) in apical membrane of the renal proximal tubule [34]. Additional studies have reported that CTS may play an important role in regulation of several pathways, including renal sodium handling and blood pressure regulation through the activation of a Src-EGFR (Epithelial Growth factor receptor) signaling cascade via caveolar Na/K ATPase [35].

## 5. Marinobufagenin and Fibrosis

It has recently been demonstrated that MBG in concentrations which are insufficient to block the pumping mechanism of the Na/K-ATPase initiates pro-fibrotic signaling by binding to the Na/K-ATPase and activating Src (sarcoma; proto-oncogene tyrosine-protein kinase) and EGFR (epidermal growth factor receptor) signaling, resulting in degradation of Fli-1 (negative nuclear regulator of the *procollagen-1* gene) in the myocardium and induction of collagen-1 synthesis [36]. Cardiac fibrosis was observed in rats administered with MBG by osmotic minipumps, and in a rat models of uremic cardiomyopathy, in which endogenous MBG concentrations were concurrently elevated [37]. High-salt diet increased TGFβ_1_ and subsequent fibrosis in the heart and kidney in both normotensive and hypertensive rats [38]. These results suggest that excessive salt intake may be an important direct pathogenic factor for cardiovascular disease. Both clinical and experimental evidence support the development of salt-induced hypertrophy of the arterial wall in the absence of arterial pressure changes [39,40]. 

In a study performed in normotensive rats, Fedorova et al. demonstrated that high salt intake stimulates MBG production and tissue remodeling in heart and kidney, without affecting BP [41]. In another study, the same authors demonstrated that MBG is essential for the development of aortic fibrosis due to high salt intake. However, immunization against MBG abrogated only the pro-fibrotic effects of a high salt diet without affecting the blood pressure [42]. High salt-intake have been also shown to paradoxically increase the tissue renin-angiotensin system activation in Dahl salt-sensitive *rats*. It was documented that such an increase of tissue angiotensin II stimulates adrenocortical MBG production in this *rat* model. Moreover, AT1 receptor blocker losartan prevented stimulation of MBG biosynthesis both in vivo and in vitro [32]. A strong relationship between high salt intake, activation of the renin-angiotensin system and pro-fibrotic signaling has been demonstrated in this study leading to the damage of cardiovascular and renal tissues. Administration of a highly specific monoclonal antibody against MBG in vivo reduced aortic fibrosis and restored aortic relaxation in animals after prolonged high salt intake. The observed changes in vascular wall morphology in the absence of hemodynamic changes indicate that possible arterial stiffening is independent of blood pressure and that the pro-fibrotic factor MBG is responsible for the development of tissue fibrosis [42].

In normotensive rats, high dietary salt intake have been associated with the activation of TGF-β signaling within the arterial wall and increased aortic stiffness in the presence of elevated levels of the Na/K-ATPase ligand MBG despite unchanged blood pressure [43]. Moreover, the *rats* exposed to a reduced salt diet after the period of high salt intake exhibited a decrease in MBG levels, downregulation of the pro-fibrotic TGF-β pathway, a decrease of aortic wall collagen content and normalization of the pulse wave velocity to control levels. The authors also demonstrated that MBG stimulates collagen production in parallel with activation of TGF-β in cultured VSMCs in vitro, in the absence of hemodynamic effects [43]. Lowering the salt intake can improve vascular elasticity and decrease the cardiovascular risk by reducing the plasma MBG concentration.

In *humans*, dietary sodium restriction has been shown to reduce urinary MBG excretion which correlated with reduction in blood pressure and aortic stiffness [10]. Most importantly, MBG excretion positively correlated with blood pressure and changes in dietary sodium intake typical for a Western diet, extending previous observations in rodents and humans fed with experimentally high-sodium diets [44].

Contrary to the findings for MBG, high doses of ouabain have been demonstrated to inhibit the TGF-β-induced fibrosis in cultured human lung fibroblasts [45,46] suggesting that different CTS may have opposing actions.

## 6. Marinobufagenin and Cardiovascular Complications

Recently we have shown that plasma marinobufagenin concentration is increased in patients with advanced chronic kidney disease irrespective of their blood pressure [47]. Moreover, the higher the plasma MBG concentration the worse the survival was in this population. Recent data from the African-PREDICT study showed that both high salt intake and elevated plasma MBG concentration were correlated with increased stiffness of large arteries measured by pulse-wave velocity [48,49]. Left ventricular mass is positively and independently associated with marinobufagenin urinary excretion in young healthy adults as well [50]. As these morphological changes also correlated with blood pressure it is not possible to differentiate the direct effects of dietary salt and MBG from the blood pressure-dependent effects. The possibility to diminish or at least postpone arterial stiffness or heart hypertrophy by simple dietary adjustments seems to be very attractive. Experimental data support such a possibility: in normotensive *rats*, low sodium diet resulted in less arterial stiffness, less vascular TGF-β-dependent fibrosis and lower plasma MBG concentration without affecting blood pressure [43]. However, as always, one has to remember that too deep an intervention also has negative effects. In an experimental study both high and low sodium diet resulted in lower nephron number and hypertension in *rat* offspring [51].

The magnitude of systolic blood pressure (SBP) response to acute change in dietary NaCl intake, the salt-sensitivity of blood pressure, increases with advancing age [4]. Specific determinants of the greater blood pressure responsiveness to dietary NaCl observed in older subjects remain to be identified. It has been proposed that salt ingestion results in an increase in plasma volume and natriuresis. It has been postulated for some time that endogenous substances are stimulated by increased Na intake and increase natriuresis by inhibiting renal tubular Na exchangers to lower the renal reabsorption of filtered sodium. The age-associated differences in circulating endogenous Na/K-ATPase inhibitors may be implicated in the age-associated increase in SBP and increased salt sensitivity of SBP in the elderly. Anderson et al. were the first to demonstrate in normotensive *humans* that following a change from a low to a high salt diet, a sustained increase in MBG synthesis occurs, and renal fractional sodium elimination increases and correlates directly with increased urinary MBG excretion. In contrast to the sustained increase in MBG synthesis on high salt diet, ouabain levels in these subjects increased only transiently [52].

## 7. Endogenous Ouabain and Other CTS

Ouabain is another cardiotonic steroid demonstrated in *human* and animal plasma. In *humans* it does not increase sodium excretion, but it does have a role in the adaptation to both sodium depletion and loading. Although a few studies have shown that high salt loading in normotensive *rats* stimulates the release of ouabain, other experiments performed in *dogs*, *rats* and *humans* did not confirm these findings. In 180 patients with untreated hypertension, plasma levels of endogenous ouabain did not change during 2 weeks of salt loading, but increased following 2 weeks of sodium depletion [53]. Recent studies indicate that endogenous ouabain might act as a central mediator of salt - sensitive hypertension. In Dahl salt-sensitive *rats*, an important interaction seems to occur between brain and peripheral cardiotonic steroids. After acute or chronic salt-loading, a transient increase in circulating endogenous ouabain precedes a sustained increase in circulating marinobufagenin concentration [54]. This observation led to the postulate that endogenous ouabain acts as a neurohormone, triggering release of MBG, which in turn increases in cardiac contractility, peripheral vasoconstriction and natriuresis by inhibiting the Na/K-ATPase. More recently it was demonstrated that, similar to observations in Dahl salt-sensitive *rats*, normotensive *humans* on increased salt intake exhibit a transient increase in urinary endogenous ouabain excretion, which precedes a more sustained increase in renal MBG excretion [52].

Experiments in Milan hypertensive *rats*, which carry a mutation in the cytoskeletal protein adducin gene and exhibit increased circulating levels of endogenous ouabain, administration of the digoxin derivative rostofuroxin antagonized the effects of ouabain and lowered blood pressure [55]. The experimental data are promising and led to a clinical trial aimed to show the hypotensive effects of rostofuroxin in *humans*. The results in *humans*, however, could not demonstrate the blood pressure lowering effects after rostafuroxin administration [56].

There is substantial uncertainty as to whether the “endogenous ouabain” is indeed identical with the plant derived ouabain [57]. Although adrenals are supposed to be the source of the endogenous ouabain, the details of the adrenal biosynthetic pathway remain to be defined. A large portion of the data supporting the presence of “endogenous ouabain” is based on immunodetection. Cross-reactivity with similar compounds is an important issue in these methods. Some authors; however, failed to detect any measurable amount of true ouabain using state-of-the-art mass spectrometry [58]. This suggests that the “endogenous ouabain” may differ slightly from the plant ouabain. Further research is definitely needed in order to determine the exact structure of the compound referred to as “endogenous ouabain”. Oubain-like immunoreactivity has been localized mainly to neuronal cells, especially hypothalamus [59]. In contrast, marinobufagenin immunoreactivity has been detected primarily in adrenals. It has been hypothesized that endogenous ouabain in the central nervous system responds to increased sodium load and increases sympathetic nervous activity resulting in hypertension [59].

Other CTS have been identified in mammalian tissues: marinobufotoxin [60], telocinobufagin [61], digoxin [62]. It is not known whether different CTS have different roles or are they different metabolites of a single active compound.

## 8. Summary

High dietary salt intake is a cause of elevated blood pressure and cardiovascular risk. However, it was demonstrated that even if the blood pressure did not increase on high salt diet, organ damage may still occur. Both effects are mediated (among other mechanisms) by endogenous digitalis-like cardiotonic steroids (Figure 2). They are released in order to maintain body sodium and act on the NaK-ATPase not only blocking the pumping mechanism but also triggering cellular responses leading to fibrosis. 

## 9. Future Perspectives

Interfering with this pathway may present a new therapeutic target for treating hypertension and cardiovascular disease. Much work is needed before drug development is possible. Antibodies that bind cardiotonic steroids are not useful for long-term treatment of hypertension and cardiovascular events, although they could be useful in short-term situations like preeclampsia. Exact molecular mechanisms in CTS biosynthesis and their regulation will be studied further. Finding a way to influence differently the Na/K-ATPase blocking and signaling functions would be a major step forward in developing new medications in this pathway.

## Figures and Tables

**Figure 1 ijms-20-00590-f001:**
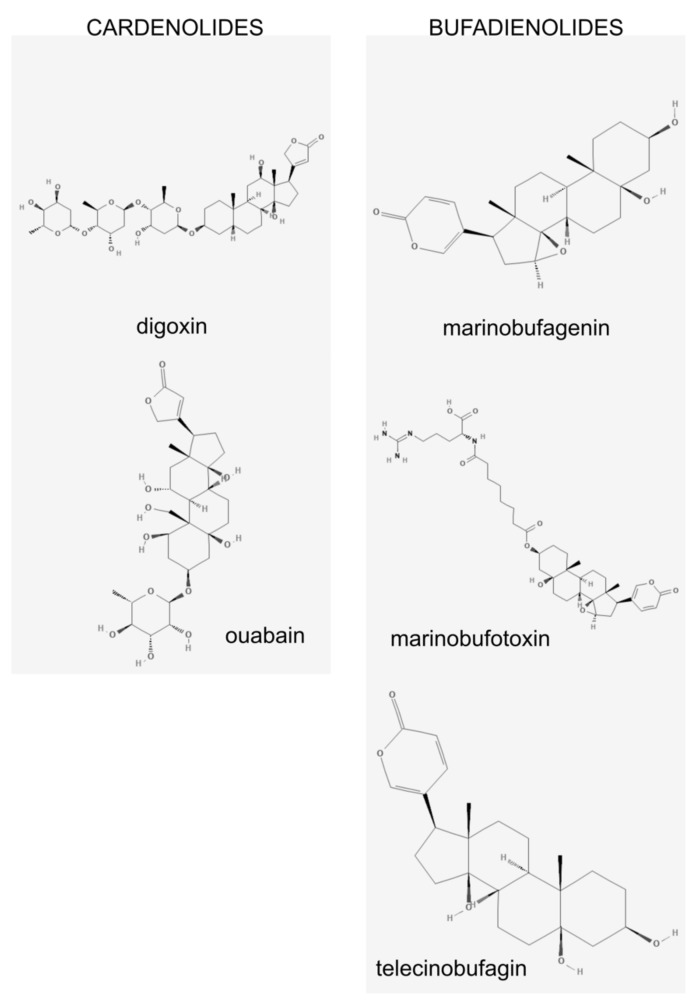
Chemical structure of cardiotonic steroids.

**Figure 2 ijms-20-00590-f002:**
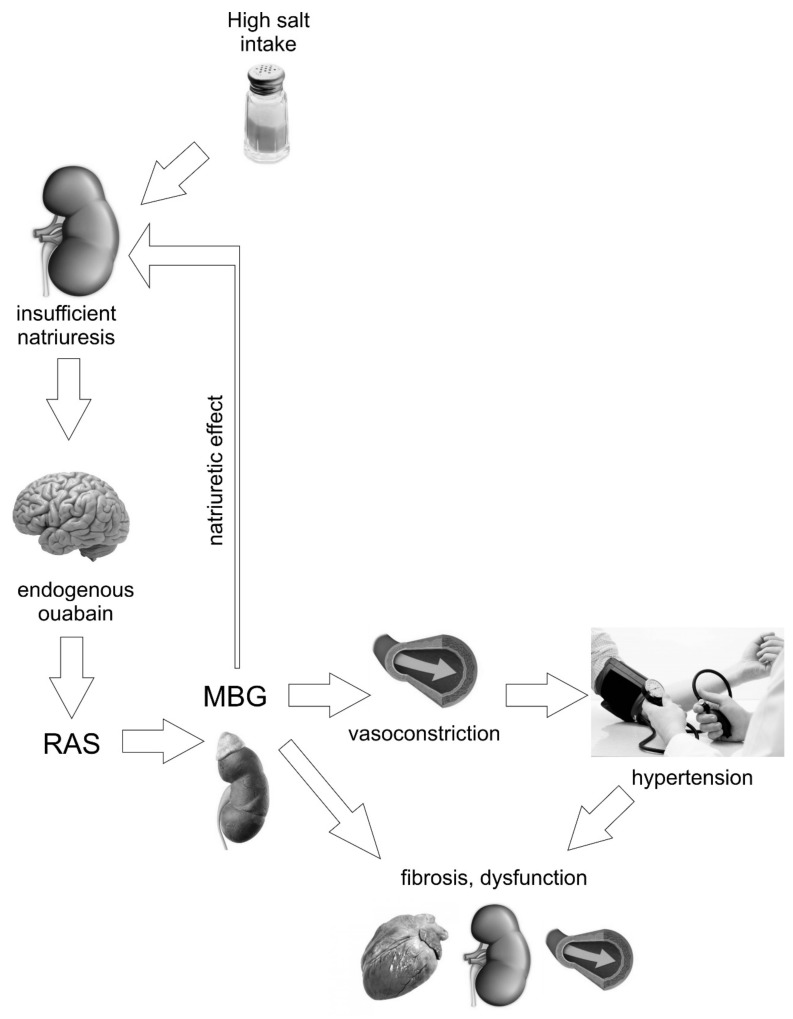
A possible mechanism of salt-induced hypertension and organ damage in humans. NaCl loading stimulates brain endogenous ouabain. Endogenous ouabain in the brain activates the local renin-angiotensin system (RAS) as well as sympathetic nervous system (SNS). These actions stimulate renin-angiotensin system in adrenal cortex and release of adrenocortical marinobufagenin (MBG). MBG is secreted in order to facilitate natriuresis, but at the same time MBG induces vasoconstriction which increases blood pressure and promotes fibrosis leading to permanent heart, kidney, and arterial damage and dysfunction.

**Table 1 ijms-20-00590-t001:** Inhibition constant (K_i_) of the Na-K-ATPase isozymes [22,23].

Isozyme	Ouabain Inhibitionin RatsK_i_, M	Ouabain Inhibitionin HumansK_i_, M
α1β1	4.3 × 10^−5^	1.3 × 10^−8^
α2β1	1.7 × 10^−7^	3.2 × 10^−8^
α2β2	1.5 × 10^−7^	
α3β1	3.1 × 10^−8^	1.7 × 10^−8^
α3β2	4.7 × 10^−8^

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
