# Peer review of "Cardiotonic Steroids—A Possible Link Between High-Salt Diet and Organ Damage"

_ijms, 2019, doi:10.3390/ijms20030590_

Reviewer 1 Report

The manuscript reviewed controversial substances, digitalis-like factors (DLFs) and proposed them as stimulators for fibrosis. Following will be addressed before publishing.

1. The text may need subtitles to follow the authors’ story more easily.

2. The controversial points were not well presented. For example, the reference indicating the absence of ouabain-like substance may need much attention. A contamination from food plants?

3. A table of DLFs including structure and plasma concentration may be helpful.

4. As the authors excel in the field of MBG, much more information on this field will be welcome. Actually, some investigators claim that brain DLFs and their effects in the brain on hypertension is more important in view of the little plasma DFLs change. ‘The central mechanism underlying hypertension: a review of the roles of sodium ions, epithelial sodium channels, the renin-angiotensin-aldosterone system, oxidative stress and endogenous digitalis in the brain. Hypertens Res. 2011 Nov;34(11):1147-60. doi: 10.1038/hr.2011.105. Epub 2011 Aug 4.` The toxin form of MBG is also present. ‘Isolation of marinobufotoxin from the supernatant of cultured PC12 cells. Clin Exp Pharmacol Physiol. 2011 May;38(5):334-7. doi: 10.1111/j.1440-1681.2011.05512.x.’

5. The parallelism with endogenous morphine-like substances, endorphin will be interesting. What will be the breakthrough of your field to match the endorphin’s. Any gene-modified animals will be helpful? Please highlight the future perspectives.

Author Response

Reviewer 1

The manuscript reviewed controversial substances, digitalis-like factors (DLFs) and proposed them as stimulators for fibrosis. Following will be addressed before publishing.

1. The text may need subtitles to follow the authors’ story more easily.

Subtitles have been added

2. The controversial points were not well presented. For example, the reference indicating the absence of ouabain-like substance may need much attention. A contamination from food plants?

We rephrased the point on ouabain detection. The structure of the ouabain-like substance is controversial – whether it is indeed ouabain or a slightly different but similar compound.

3. A table of DLFs including structure and plasma concentration may be helpful.

We added a figure depicting DLFs structure

4. As the authors excel in the field of MBG, much more information on this field will be welcome. Actually, some investigators claim that brain DLFs and their effects in the brain on hypertension is more important in view of the little plasma DFLs change. ‘The central mechanism underlying hypertension: a review of the roles of sodium ions, epithelial sodium channels, the renin-angiotensin-aldosterone system, oxidative stress and endogenous digitalis in the brain. Hypertens Res. 2011 Nov;34(11):1147-60. doi: 10.1038/hr.2011.105. Epub 2011 Aug 4.` The toxin form of MBG is also present. ‘Isolation of marinobufotoxin from the supernatant of cultured PC12 cells. Clin Exp Pharmacol Physiol. 2011 May;38(5):334-7. doi: 10.1111/j.1440-1681.2011.05512.x.’

We added the information regarding brain DLFs and other DLFs that has been detected in mammalian tissues.

5. The parallelism with endogenous morphine-like substances, endorphin will be interesting. What will be the breakthrough of your field to match the endorphin’s. Any gene-modified animals will be helpful? Please highlight the future perspectives.

We highlighted the future perspectives.

Reviewer 2 Report

Paczula A et al Cardiotonic steroids – a possible link …”

 This review focus on the recent data indicating an implication of high salt diet and augmented production of cardiotonic steroids (CTS) (mainly marinobufagenin, MBG) in cardiovascular and renal fibrosis.

1.     Recent studies demonstrated that abnormal renal handling of salt and osmotically obliged water can not be considered as the only mechanisms of the pathogenesis of slat-sensitive disorders. These mechanisms are considered in several comprehensive reviews [1-3] and should be mentioned in the revised manuscript.

2.     Several important statements are not supported by related referenced (line 50, line 146, line 179-187).

3.     Lines 69-70: “The cardenolides have been determined to have a predilection for the α2 and a3 isoforms, whereas the bufadienolides act primarily on the α1 isoform”. This key statement should be supported by comprehensive analysis of the available data on the affinity of a1, a2 and a3 isoforms for cardenolides and bufadienolides presented as a Table. Another Table showing the content of endogenous cardenolides and bufadienolides should be also added to the review. Keeping in mind the drastic differences in the affinity of a1 isoform data obtained in rodents and human should be separated within these Tables. To simplify this task, Table presented by Khalaf and co-workers in their recent review [4] might be used in this section. This Table shows that at least in rodents CTS can not be considered as endogenous natriuretic hormone affecting renal function via inhibition of the a1-Na+,K+-ATPase. This step is shown in Fig. 1.

4.     Lines 71-72 “in rats the α1 isoform is sensitive to ouabain, while in humans is not” This misleading sentence is probably wrong

5.     Lines 113-115. “High salt diet is also associated with the stimulation of TGFβ and subsequent fibrosis in the cardiovascular tissue in young normotensive rats”. This statement is not obvious and should be explained with more details.

6.     Recent studies demonstrated that high doses of cardenolides inhibit TGF-b-induced fibrosis in cultured human lung fibroblasts via [Na+]i/[K+]i-mediated signaling via augmented expression of COX-2 and downregulation of TGF-b receptor [5;6]. These papers should be mentioned in the revised manuscript.

References

  1.  Orlov SN, Mongin AA. Salt sensing mechanisms in blood pressure regulation and hypertension. Am J Physiol Heart Circ Physiol 2007; 293: H2039-H2053.

  2.  Oberleithner H, Kusche-Vihrog K, Schillers H. Endothelial cells as vascular salt sensor. Kidney Int 2010; 77: 490-494.

  3.  Titze J. Sodium balance is not just a renal affair. Curr Opin Nephrol Hypert 2014; 23: 101-105.

  4.  Khalaf FK, Dube P, Mohamed A et al. Cardiotonic steroids and the sodium trade balance: new insights into trade-off mechanisms mediated by the Na+,K+-ATPase. Int J Mol Sci 2018; 19: doi: 10.3390/ijms19092576.

  5.  La J et al. Regulation of myofibroblast differentiation by cardiac glycosides. Am J Physiol Lung Cell Mol Physiol 2016; 310: L815-L823.

  6.  La J et al. Downregulation of TGF-beta receptor-2 expression and signaling through inhibition of Na/K-ATPase. PLoS One 2016; 11: e0168363.

 Author Response

Reviewer 2

This review focus on the recent data indicating an implication of high salt diet and augmented production of cardiotonic steroids (CTS) (mainly marinobufagenin, MBG) in cardiovascular and renal fibrosis.

1.     Recent studies demonstrated that abnormal renal handling of salt and osmotically obliged water can not be considered as the only mechanisms of the pathogenesis of slat-sensitive disorders. These mechanisms are considered in several comprehensive reviews [1-3] and should be mentioned in the revised manuscript.

Following the reviewers suggestion we added this information.

2.     Several important statements are not supported by related referenced (line 50, line 146, line 179-187).

We added the missing references.

3.     Lines 69-70: “The cardenolides have been determined to have a predilection for the α2 and a3 isoforms, whereas the bufadienolides act primarily on the α1 isoform”. This key statement should be supported by comprehensive analysis of the available data on the affinity of a1, a2 and a3 isoforms for cardenolides and bufadienolides presented as a Table. Another Table showing the content of endogenous cardenolides and bufadienolides should be also added to the review. Keeping in mind the drastic differences in the affinity of a1 isoform data obtained in rodents and human should be separated within these Tables. To simplify this task, Table presented by Khalaf and co-workers in their recent review [4] might be used in this section. This Table shows that at least in rodents CTS can not be considered as endogenous natriuretic hormone affecting renal function via inhibition of the a1-Na+,K+-ATPase. This step is shown in Fig. 1.

We added a table and more information in the text regarding the affinity of the Na/K-ATPase and concentrations of CTS reported in humans and rodents.

4.     Lines 71-72 “in rats the α1 isoform is sensitive to ouabain, while in humans is not” This misleading sentence is probably wrong

We removed the incorrect sentence

5.     Lines 113-115. “High salt diet is also associated with the stimulation of TGFβ and subsequent fibrosis in the cardiovascular tissue in young normotensive rats”. This statement is not obvious and should be explained with more details.

We added more information and a reference to the statement.

6.     Recent studies demonstrated that high doses of cardenolides inhibit TGF-b-induced fibrosis in cultured human lung fibroblasts via [Na+]i/[K+]i-mediated signaling via augmented expression of COX-2 and downregulation of TGF-b receptor [5;6]. These papers should be mentioned in the revised manuscript.

We added this information

References

  1.  Orlov SN, Mongin AA. Salt sensing mechanisms in blood pressure regulation and hypertension. Am J Physiol Heart Circ Physiol 2007; 293: H2039-H2053.

  2.  Oberleithner H, Kusche-Vihrog K, Schillers H. Endothelial cells as vascular salt sensor. Kidney Int 2010; 77: 490-494.

  3.  Titze J. Sodium balance is not just a renal affair. Curr Opin Nephrol Hypert 2014; 23: 101-105.

  4.  Khalaf FK, Dube P, Mohamed A et al. Cardiotonic steroids and the sodium trade balance: new insights into trade-off mechanisms mediated by the Na+,K+-ATPase. Int J Mol Sci 2018; 19: doi: 10.3390/ijms19092576.

  5.  La J et al. Regulation of myofibroblast differentiation by cardiac glycosides. Am J Physiol Lung Cell Mol Physiol 2016; 310: L815-L823.

  6.  La J et al. Downregulation of TGF-beta receptor-2 expression and signaling through inhibition of Na/K-ATPase. PLoS One 2016; 11: e0168363.

Round  2

Reviewer 1 Report

The manuscript has been improved. I have no further comments.

Reviewer 2 Report

All my comments are considered in the revised manuscript

This manuscript is a resubmission of an earlier submission. The following is a list of the peer review reports and author responses from that submission.